# Age-Related Differences in Kinematics, Kinetics, and Muscle Synergy Patterns Following a Sudden Gait Perturbation: Changes in Movement Strategies and Implications for Fall Prevention Rehabilitation

**Woohyoung Jeon** [1],* **, Ahmed Ramadan** [2] **, Jill Whitall** [3] **, Nesreen Alissa** [3] **and Kelly Westlake** [3]

1  Department of Health & Kinesiology, University of Texas at Tyler, Tyler, TX 75701, USA
2  Department of Biomedical Engineering, University of Minnesota Twin Cities, Minneapolis, MN 55455, USA;
   aramadan@umn.edu
3  Department of Physical Therapy & Rehabilitation Science, University of Maryland School of Medicine,
   Baltimore, MD 21201, USA; jwhitall@som.umaryland.edu (J.W.); nesreen.alissa@som.umaryland.edu (N.A.);
   kwestlake@som.umaryland.edu (K.W.)
*  Correspondence: wjeon@uttyler.edu; Tel.: +1-(903)566-7042

**Abstract:** Falls in older adults are leading causes of fatal and non-fatal injuries, negatively impacting quality of life among those in this demographic. Most elderly falls occur due to unrecoverable limb collapse during balance control in the single-limb support (SLS) phase. To understand why older adults are more susceptible to falls than younger adults, we investigated age-related differences in lower limb kinematics, kinetics, and muscle synergy patterns during SLS, as well as their relationship to postural control strategies. Thirteen older and thirteen younger healthy adults were compared during the SLS phase of balance recovery following an unexpected surface drop perturbation. Compared to younger adults, older adults demonstrated (1) greater trunk flexion, (2) increased hip extension torque and reduced hip abduction torque of the perturbed leg, and (3) higher postural sway. Trunk flexion was correlated with a delayed latency to the start of lateral-to-medial displacement of center of mass from the perturbation onset. The group-specific muscle synergy revealed that older adults exhibited prominent activation of the hip extensors, while younger adults showed prominent activation of the hip abductors. These findings provide insights into targeted balance rehabilitation and indicate ways to improve postural stability and reduce falls in older adults.

**Keywords:** older adults; muscle synergy; movement strategy; balance control

## 1. Introduction

Falls are common among older adults over 65 years old, and approximately one out of five falls result in severe fatal and non-fatal injuries [1], such as hip fractures and traumatic brain injuries. The rate of falls leading to these severe injuries increases with age [2,3]. Therefore, the ability to successfully maintain postural stability and recover balance is important for preventing falls in older adults.

A progressive decline in balance control is a natural part of the aging process. Older adults often encounter difficulties in maintaining postural stability when recovering from external perturbations. Previous studies have demonstrated that age-related changes in neuromuscular control, muscle weakness, and decreased flexibility lead to altered muscle activation patterns during balance recovery while standing and walking. For example, compared to younger adults, older adults demonstrate less modulation of spinal reflexes and more co-contraction at the ankle joint for balance control during standing on a compliant surface [4]. A decrease in the efficiency of plantar flexion push-off power during standing balance control in response to lateral surface perturbations has also been observed in older adults [5]. While walking, neuromuscular control tends to become more simplistic

with age, leading to reduced variability and complexity in the available muscle synergy options [6]. In addition, age-related reductions in lower limb force generation and reduced flexibility lead to altered muscle activation patterns during dynamic balance control [5,7]. These factors collectively contribute to impaired postural stability when older adults need to respond to balance challenges.

Most outdoor falls in older adults occur when they encounter challenges while walking on different types of surfaces, such as uneven and slippery terrain [8,9]. The ability to successfully recover balance following unexpected ground perturbations depends on how well the forward and downward falling momentum is decelerated in the anterior-posterior (AP) and vertical directions while simultaneously maintaining postural stability in the medio-lateral (ML) direction [10–13]. However, compared to younger adults, older adults exhibit impaired and less efficient balance control across the joints of their lower limbs.

For example, older adults have been shown to demonstrate a relatively greater magnitude of electromyography (EMG) activation in ankle plantar flexor despite having no age-related differences in plantar flexion torque during ankle joint push-off [5]. At the knee joint level, older adults exhibit reduced knee extensor eccentric work and EMG burst duration during the single-limb support (SLS) phase of balance recovery, which is associated with reduced absorption of falling momentum and forward momentum control [14]. In addition, older adults display diminished hip abduction torque during protective stepping for balance recovery from lateral perturbations [15]. This reduction indicates compromised control of interlimb weight transfer at the hip joint, which is crucial for maintaining a stable relationship between the body's center of mass (CoM) and base of support (BoS). As a result, postural instability in the ML direction increases.

Age-related declines in dynamic balance control in the lower extremities are associated with changes in postural control strategies in older adults while walking. The impairment in ankle joint force control which becomes more observable with age leads to an altered movement strategy for balance control, redistributing push-off propulsive force generation to more proximal knee and hip joints while walking in older adults [16]. Additionally, older adults demonstrate ineffective inter-joint coordination, characterized by the increased coactivation of agonist and antagonist muscles at the ankle and knee joints while walking [17], and increased joint kinematic variability at the ankle, knee, and hip joints during walking with lateral balance perturbations [18]. These age-related changes in postural control strategies during dynamic balance control may raise potential risk of falls in older adults.

From a biomechanical perspective, falls in older adults result from what has been described as an unrecoverable limb collapse of the perturbed leg during single-limb support (SLS) following ground balance challenges [19,20]. Once the protective (compensatory) step of swinging the trailing leg is initiated, the SLS of the perturbed leg is critical, as it is the initial defense against limb collapse immediately after ground perturbation [19]. During the perturbed leg SLS phase, the falling and forward momentum caused by ground perturbation are decelerated, and the swinging leg prepares for the protective (compensatory) step. Thus, analyzing the kinematics, kinetics, and muscle synergy patterns during the SLS phase could help identify the age-specific movement characteristics and/or strategies for balance control and reveal why older adults are more prone to fall from the same level of balance perturbation that their younger counterparts can successfully recover from.

The purpose of this study was to investigate age-related differences in kinematics, kinetics, and muscle synergy patterns during the SLS phase following unexpected surface drop perturbations. Additionally, the study aimed to explore the relationship of these factors to postural control strategies for balance recovery. The results of this investigation will provide valuable insights regarding lower limb strengthening exercises which aim to improve dynamic balance control and reduce fall risk among older adults.

## 2. Materials and Methods

### 2.1. Participants

A total of thirteen healthy younger adults ($24 \pm 3$ years; 6 females, 7 males) and thirteen healthy older adults ($77 \pm 8$ years; 6 females, 7 males) participated in this study. The participants were recruited through a weekly department newsletter, online campus advertisements, and the "ResearchMatch" program website. Older adults were defined as individuals over 65 years of age, while younger adults were considered to be those between 18 and 30 years old. The physical activity level of the participants (the number of days and hours spent walking and doing physical activities per week) was assessed using the International Physical Activity Questionnaire [21]. To be eligible for inclusion in the study, participants had to meet the following criteria: (1) be able to walk 10 m without an assistive device and (2) have a "moderate" or high physical activity level, which is defined as (a) 3 or more days of high-intensity activity for at least 20 min per day or (b) 5 or more days of moderate-intensity activity and/or walking for at least 30 min per day or (c) 5 or more days of any combination of walking, moderate-intensity, or high-intensity activities as long as they partake in physical activity for a minimum total of at least 600 min/week. The participants' characteristics are provided in Table 1.

**Table 1.** Anthropometrics and spatiotemporal gait characteristics of study participants across age groups. Values are presented as mean $\pm$ standard deviation. Single-stance duration is equal to the swing time of the opposite foot. H–H (heel-to-heel) base of support is "base width", which is the vertical distance from the heel center of one footprint to the line of progression formed by two footprints of the opposite foot. * represents a significant difference between the two groups.

| Characteristics | Older Adults ($n$ = 13) | Younger Adults ($n$ = 13) | $p$-Value |
|---|---|---|---|
| Anthropometric | | | |
| Age (years) | $77 \pm 7$ | $22 \pm 3$ | |
| Sex (female/male) | 6/7 | 6/7 | |
| Height (cm) | $170.8 \pm 5.32$ | $173.0 \pm 7.77$ | 0.75 |
| Weight (kg) | $74.33 \pm 11.21$ | $73.74 \pm 16.55$ | 0.74 |
| BMI (kg/m$^2$) | $23.02 \pm 0.46$ | $22.33 \pm 3.02$ | 0.19 |
| Gait | | | |
| Gait Speed (cm/s) | $111.23 \pm 21.13$ | $122.14 \pm 14.30$ | 0.25 |
| Gait Initiation step length (cm) | $51.33 \pm 5.33$ | $55.90 \pm 5.40$ | 0.10 |
| Step length (cm) | $59.24 \pm 5.96$ | $69.32 \pm 4.10$ | 0.03 * |
| Single stance duration (s) | $0.45 \pm 0.12$ | $0.44 \pm 0.02$ | 0.49 |
| H–H Base of Support (cm) | $10.29 \pm 3.41$ | $11.89 \pm 3.16$ | 0.15 |

Participants were excluded from this study if they had (1) deficits or disorders that could affect balance control; (2) history of dizziness and imbalance; (3) history of neurological (e.g., Parkinson's disease, Alzheimer's disease, stroke, visual and/or vestibular impairment), musculoskeletal disorders or any other systemic disorders; and/or (4) a Body Mass Index (BMI) result within the overweight and obesity range (i.e., a BMI result higher than $25 \, \text{kg m}^{-2}$). Testing was performed in the morning hours (9 am to 11 am) to prevent the effect of daytime sleepiness on balance control (Forsman et al., 2007). All procedures conducted for the study were approved by the Institutional Review Board at the University of Maryland (protocol code: HP-00093655 and 21 January 2021) and were conducted in accordance with the Declaration of Helsinki.

All participants provided written informed consent prior to participating in the study.

### 2.2. Data Collection

A Noraxon TeleMyo wireless EMG System (Noraxon, Scottsdale, AZ, USA) was used to detect muscle activity signals. Noraxon adhesive pre-gelled Ag/AgCl surface EMG electrodes ($40 \times 21$ mm, inter-electrode distance: 20 mm) were bilaterally placed on the tibialis anterior (TA), medial gastrocnemius (mGas), rectus femoris (RF), biceps femoris

(BF), gluteus medius (Gmed), and erector spinae (ES). The electrode positioning followed the guideline provided by SENIAM [22]. Thirty-nine reflective markers were placed on the body in accordance with the standard Plug-In-Gait full-body model (Vicon Nexsus 2.12, VICON Motion Systems Ltd., Yarnton, UK). A 10-camera motion capture system (VICON Motion Systems Ltd., UK) was used to record body kinematics.

The surface drop perturbation was applied at heel strike during gait initiation to standardize the timing of the perturbation and normalize kinematic responses to the same phase of the gait cycle. Prior to the start of testing, spatiotemporal gait data (usual walking speed, step length) during unperturbed gait were collected from three trials using the GaitRite mat (Protokinetics, Havertown, PA, USA). Step length was used to determine the starting position for each participant, ensuring that the leading leg's heel strike occurred at the center location on the unexpected surface drop platform. As our focus was on capturing participants' initial naïve response to the ground walking balance challenge, only the first exposure to the surface drop balance perturbation was measured.

During testing, participants were fitted with a safety harness that was connected to an overhead pulley system with minimal resistance. The length of the rope was adjusted to match each participant's height, allowing them to walk freely. Participants were instructed to walk at their normal pace along a 30′ (9.14 m) walkway (U.S. Provisional Pat. Ser. No. 62/949,184). The walkway was modular, and the first 3′ module was located before the surface drop module. Spring hinges held the drop panel (44″W × 24″H) along the boardwalk in place until activated via heel strike of the leading limb (Figure 1). The instructions provided to the participants were as follows: "Walk to the end of the walkway at your normal walking pace. Your balance may or may not be challenged. If your balance is disturbed, react naturally and continue walking to the end of the walkway".

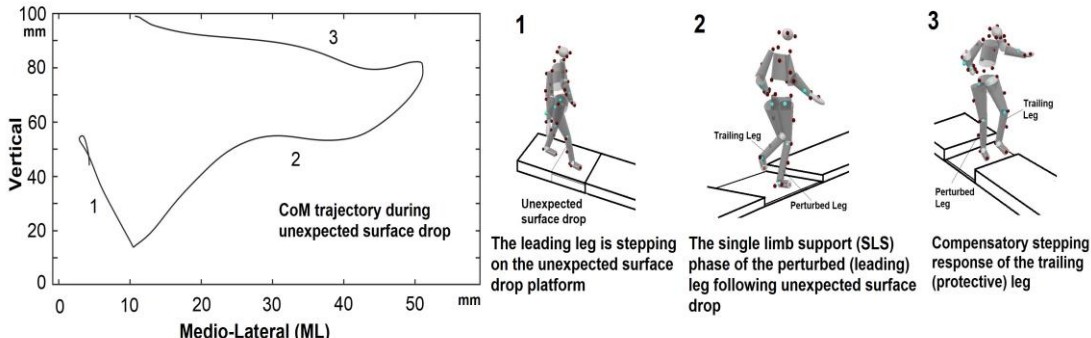

**Figure 1.** Center of mass trajectory during the experiment and experimental setup for the unexpected surface drop perturbation. The surface drop was triggered by the heel strike of the leading leg during gait initiation. Spring hinges held a panel (shaded in grey) in place until it is activated by a leading (perturbed) leg stepping. The trailing (protective) leg stepped onto the firm surface of the platform (compensatory stepping). (1) A surface drop occurred when participants' heels struck their leading leg during gait initiation. (2) The single-limb support (SLS) phase occurs when the leading (perturbed) leg is in contact with the ground. The body's kinematics, kinetics, and muscle synergy patterns were measured in this phase. (3) The trailing (protective) leg performs a compensatory step to safely land on the firm surface of the platform during the SLS.

### 2.3. Data Processing

The raw surface EMG response to the perturbation was filtered with a 20–450 Hz band-pass filter. Subsequently, the EMG data were high-pass filtered at 35 Hz, and a 2nd order Butterworth low-pass filter with a 40 Hz cutoff was applied as a digital smoothing algorithm after full-wave rectification [14,23]. EMG data were sampled at 1500 Hz, and kinematic data were sampled at 150 Hz. EMG, kinematics, and kinetics data were analyzed in Matlab 2022b (Matworks Inc., Natick, MA, USA).

### 2.3.1. Kinematic and Kinetic Data

The body's center of mass (CoM) trajectory, angular displacement, and torque of the ankle and hip joints were calculated using the Plug-in Gait dynamic pipeline (Vicon, Oxford Metrics, Yarnton, UK). Trunk flexion angle was defined as the angle between the thorax and the laboratory coordinate system. Postural sway (body oscillation) during the SLS phase of the perturbed leg was calculated to examine postural stability. To quantify postural sway, the standard deviations (SD) of CoM acceleration (SDCoMAccel) in the AP, ML, and vertical directions were computed. Kinematic and kinetic data were Butterworth low-pass filtered at 6 Hz and 25 Hz, respectively [24].

### 2.3.2. EMG

The surface EMG response to perturbation was filtered using a 20–450 Hz band-pass filter. A second-order Butterworth low-pass filter with 20 Hz cutoff was applied as a digital smoothing algorithm after full-wave rectification [25].

### 2.3.3. Muscle Synergy

To characterize the muscle activation patterns for movement strategies, we performed muscle synergy analysis using non-negative matrix factorization. Since we only measured the first exposure (a single trial) to the balance challenge on the compliant (foam) surface, the synergies were extracted from this single trial.

Extraction of muscle synergy: Nonnegative matrix factorization (NNMF)—using a multiplicative iterative algorithm, muscle synergies (muscle weighting and synergy activation) were extracted from the matrix [26,27]. Muscle weighting (W) is the spatial component that implies the relative contribution of each muscle during the movement. Synergy activation (C) is the temporal recruitment coefficient (a time-varying component), which indicates synergy recruitment over time. This transformation can be expressed as:

$$EMG_{0\ (m \times t)} = W_{(m \times n)} \cdot C_{(n \times t)} + e = EMGr_{(m \times t)} + e, \tag{1}$$

(where m = the number of muscles, t = the number of time points, n = the number of muscle synergies, e = residual error, and EMGr = reconstructed EMG matrix).

We normalized synergy activation to the maximum activation; therefore, they ranged from 0 to 1 [28]. To evaluate the similarity between $EMG_0$ and EMGr, variability accounted for (VAF) was calculated according to the following equation:

$$VAF = \left(1 - \frac{(EMG_0 - EMG_r)^2}{(EMG_0 - mean(EMG_r))^2}\right) \times 100\% \tag{2}$$

To determine the optimal number of synergies, we repeated the optimization to extract k (from 1 to the number of EMG sensors) synergies and the associated VAF. Then, the smallest k with VAF > 90% was selected [29].

A k-means clustering algorithm in Matlab 2022b was used to categorize the similar groups of muscle synergies across all participants. Then, intra-class correlation coefficient (ICC) was used to exam the internal consistency of all muscle synergies. Based on the 95% confidence interval of the ICC estimates, ICC muscle synergy values over 0.75, indicating good reliability, were categorized in the same cluster. To determine age-specific differences, ICC muscle synergy values over 0.9, indicating excellent reliability, were categorized in the same cluster [30].

### 2.4. Statistical Analysis

SPSS statistical software (IBM SPSS Statistics 25; Chicago, IL, USA) with an established a priori alpha level of 0.05 was used for all statistical analysis. A priori power analyses using G*Power were performed for sample size justification. Effect size (Cohen's d) was determined by calculating the mean difference between the two groups based on previous

research [15,31,32]. An effect size of 1.154 was detected when there was 80% power at the 0.05 alpha level (two-tailed). The Shapiro–Wilk test was performed to test the normality of the data. All kinematic and kinetic data were normally distributed.

One-way multivariate analysis of variance (MANOVA) was used to examine whether there were any significant differences between the age groups (old and young) regarding kinematics (maximum angular displacement of the perturbed leg's ankle plantarflexion and dorsiflexion, hip flexion and extension, hip adduction and abduction, trunk flexion and extension, and latency to the start of lateral-to-medial (L-to-M) displacement of CoM from the perturbation onset), kinetics (peak torques in the AP and ML directions at the ankle and hip joints of the perturbed leg, and peak torque of trunk flexion and extension), and postural sway (measured by SD of CoM acceleration, SDCoMAccel) during the SLS phase. Tukey's test was used for post hoc analyses, as well as pairwise comparisons between two groups, where indicated. Pearson's correlation (r) was used to examine the correlation between trunk flexion angular displacement and the latency to the start of L-to m displacement of CoM from the perturbation onset.

## 3. Results

All data are presented as mean $\pm$ standard deviation (SD).

### 3.1. Kinematics and Kinetics

3.1.1. Angular Displacement and Torque

At the ankle joint, there were no differences between older and younger adults in ankle dorsiflexion and plantarflexion angular displacement and torque during the SLS phase (Figure 2A,B). However, at the hip joint, older adults exhibited greater hip extension torque (F = 7.51, $p = 0.01$, $\eta^2 = 0.23$), while younger adults showed greater hip abduction torque (F = 5.65, $p = 0.03$, $\eta^2 = 0.19$, Figure 2D).

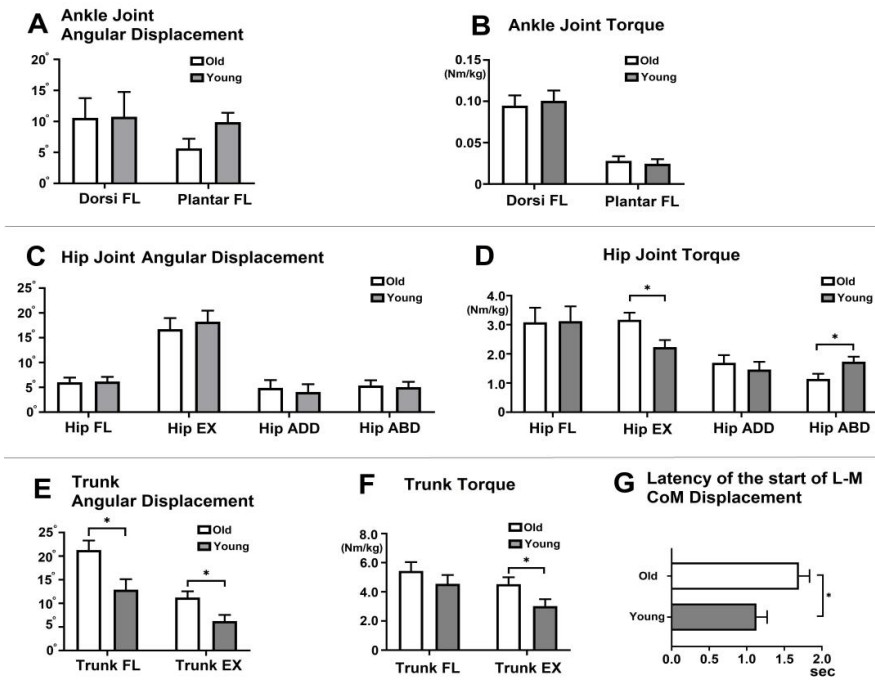

**Figure 2.** Kinematics and kinetics during the single-limb support (SLS) phase (**A**). Ankle joint angular displacement (°) (**B**). Ankle joint torque (Nm/kg) (**C**). Hip joint angular displacement (°) (**D**). Hip joint torque (Nm/kg) (**E**). Trunk angular displacement (°) (**F**). Trunk Torque (Nm/kg). (**G**) Latency to the start of L m CoM displacement during the SLS phase. In the figure, asterisks (*) indicate statistically significant differences between older and younger adults ($p < 0.05$). The error bars represent the standard deviation. FL (flexion), EX (extension), Sec (second), L m (Lateral-to-Medial), CoM (center of mass).

Greater trunk flexion (F = 8.68, $p$ = 0.01, $\eta^2$ = 0.26) and extension (F = 7.22, $p$ = 0.01, $\eta^2$ = 0.23, Figure 2E) angular displacement were observed in older adults. Additionally, older adults demonstrated greater trunk extension torque compared to younger adults (F = 5.15, $p$ = 0.03, $\eta^2$ = 0.17, Figure 2F). Older adults exhibited a delayed latency (time in seconds) to the start of L-to m displacement of the CoM from the perturbation onset compared to younger adults (F = 4.65, $p$ = 0.04, $\eta^2$ = 0.16, Figure 2G).

### 3.1.2. Relationship between Trunk Flexion and the Latency to the Start of Lateral to Medial CoM Displacement

Across all participants, there was a positive correlation between trunk flexion angular displacement (°) and latency to the start of L-to m displacement of CoM from the onset of the surface drop (r = 0.41, $p$ = 0.04, Figure 3B).

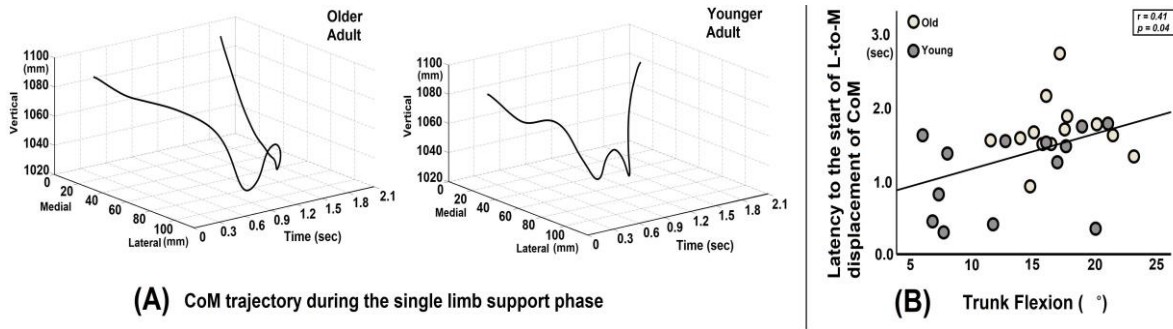

**Figure 3.** (**A**) Graph illustrating the whole-body center of mass (CoM) trajectory in the medio-lateral and vertical directions for a representative older adult and a younger adult. The older adult exhibits a delayed latency to the start of lateral-to-medial (L-to-M) displacement of CoM from the onset of the surface drop. (**B**) The correlation plot shows a positive relationship between trunk flexion angular displacement and the latency (time in seconds) to the start of L-to m displacement of the CoM from the onset of the surface drop perturbation.

### 3.1.3. Postural Sway during SLS

Postural sway, represented by standard deviation of CoM acceleration (SDCoMAccel), was greater in the vertical direction than the AP ($p$ = 0.01) and ML directions ($p$ < 0.01). Compared to younger adults, older adults demonstrated greater postural sway (SDCoMAccel) in the ML ($p$ = 0.04) and vertical direction ($p$ < 0.01) but not in AP direction ($p$ = 0.13) direction (Figure 4).

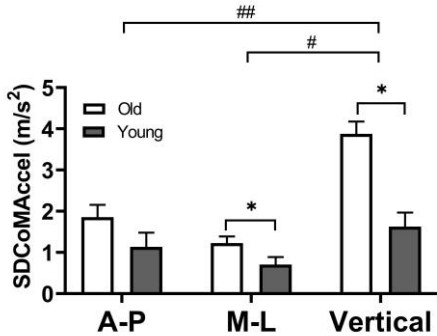

**Figure 4.** Postural sway (SD of CoM acceleration) during the single-limb support phase in the AP (anterior-posterior), ML (medio-lateral), and vertical directions. The error bars display the standard deviation. # represent a statistically significant difference between the ML and vertical directions ($p$ < 0.05). ## represent a statistically significant difference between the AP and vertical directions ($p$ < 0.05). * represent a statistically significant difference between older and younger adults ($p$ < 0.05).

### 3.2. Muscle Synergy

We identified common and group-specific muscle synergies between older and younger adults in the early and middle phases of SLS. We found a cluster showing good reliability (ICC value $\geq$ 0.75) across all participants. This common muscle synergy cluster was primarily characterized by the activation of the mGas muscle. Group-specific muscle synergies exhibited excellent reliability (ICC value $\geq$ 0.90). The older adult-specific muscle synergy clusters exhibited predominant BF muscle activation, while the younger adult-specific muscle synergy cluster exhibited predominant Gmed activation (Figure 5).

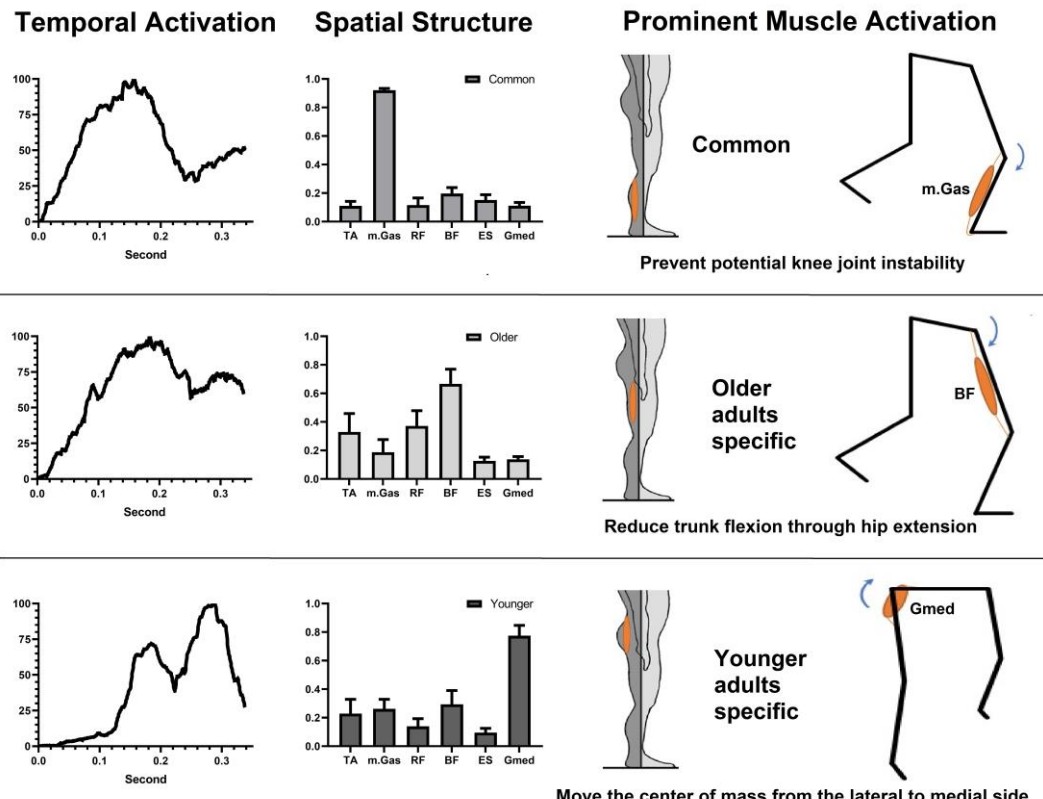

**Figure 5.** Common and specific muscle synergies in older and younger adults in the single-limb support phase.

### 4. Discussion

The purpose of this study was to investigate age-related changes in kinematics, kinetics, and muscle synergy patterns during the SLS phase following unexpected surface drop perturbations and gain insight into how these factors relate to postural control strategies for balance recovery. In comparison to younger adults, older adults demonstrated greater postural sway, trunk flexion, hip extension torque, and a delayed latency to the start of L-to m displacement of CoM from the perturbation onset, whereas younger adults displayed increased hip abduction torque during the SLS phase of balance response. Additionally, greater trunk flexion was correlated with a delayed latency to the start of L-to m displacement of CoM from the perturbation onset.

Both age groups exhibited common muscle synergy characterized by prominent activation of the mGas muscle. Regarding the age-specific muscle synergies, older adults displayed a distinct muscle synergy pattern, exhibiting prominent activation of the BF muscle, while younger adults exhibited prominent activation of the Gmed muscle during the SLS of the perturbed leg.

### 4.1. Kinematic and Kinetic Differences between Older and Younger Adults

It has been observed that older adults display greater trunk flexion during balance recovery reactions compared to younger adults [33,34]. An unexpected surface drop leads to a reflex overaction of the back muscles, resulting in greater trunk flexion [35]. This trunk flexion following an unexpected surface drop poses a challenge to the dynamic stability of forward-directed locomotion [36]. The trunk represents a significant portion of the total body mass, accounting for more than 50%, and its position above the ground is relatively high [37]. Thus, excessive trunk flexion during balance recovery negatively affects balance control, potentially increasing the risk of falls [38].

We observed that older adults exhibited greater hip extension of the leading (perturbed) leg during the SLS phase compared to younger adults. This finding suggests that older adults employ biomechanical strategies that involve increased hip extension to counteract the disadvantage of greater trunk flexion for better balance control.

From the biomechanical perspective, the greater trunk flexion observed during the single-limb stance phase may lead to higher passive tension on the hamstring muscles, and this increased tension could result in greater activation of the hamstring muscles in older adults [39]. Increased trunk flexion while walking is also associated with an increase in mechanical energy demand generated by hip extensor torque [40]. Trunk flexion alters the alignment and position of the body's CoM, shifting it forward [41]; therefore, hip extensor muscles are required to generate more torque to counterbalance the forward movement of the body's CoM and maintain postural stability [42].

Given that the increase in hip extension serves as a compensatory mechanism to counteract the anterior pelvic tilt associated with trunk flexion [42], this prominent hip extension can be considered as a movement strategy employed by older adults to improve the dynamic stability of forward-directed locomotion, which is closely related to the risk of falls. However, the higher amplitude of CoM oscillation in the vertical direction observed in older adults during the balance recovery process in the SLS phase indicates that greater trunk flexion and hip extension can negatively impact vertical balance, leading to increased postural sway in the vertical direction.

In addition, increased demand for AP balance control, resulting from the control of greater trunk flexion [43], may disrupt ML balance control during the balance recovery process. When the leading (perturbed) leg steps onto the surface drop platform, the CoM shifts towards the perturbed leg and continues moving toward the lateral side of the perturbed leg (see Figure 3A). To maintain postural stability in the ML direction, it is essential to reposition the CoM medially from the lateral side towards the inside of the base of support, which is the center of the body in the frontal plane. This L-to m displacement of the CoM is important in preventing potential loss of balance in the ML direction [44].

We found that older adults exhibited a delayed latency in initiating the L-to m displacement of CoM compared to younger adults following perturbation onset. This delayed latency was found to be positively correlated with the degree of trunk flexion. In other words, greater trunk flexion means that more time is required for the CoM to return from the lateral to medial side during the SLS phase, which is crucial for maintaining ML balance control.

This finding provides important insights into the relationship between trunk flexion control in the AP direction and hip abduction for ML balance control. Younger adults who exhibited relatively less trunk flexion during the balance recovery process demonstrated earlier initiation of L-to m CoM displacement with greater hip abduction torque when compared to older adults. This suggests that excessive trunk flexion in older adults may interfere with the timely activation of hip abduction, hindering the active control of ML balance.

### 4.2. Age-Related Differences in Muscle Synergy Patterns and Their Roles in Balance Control

4.2.1. Common Muscle Synergy

Our findings indicate that the activation of the medial gastrocnemius (mGas) is a prominent component of the observed muscle synergy in both older and younger adults during the initial phase of SLS. The mGas is a biarticular muscle, meaning it has attachments at both the ankle and knee joints. When the foot is dorsiflexed, the mGas contributes to knee flexion. This knee flexion torque created by the mGas can generate compressive shear force at the knee joint [45]. During the initial single-leg landing phase, the compressive shear force generated by mGas stabilizes the SLS balance by preventing anterior tibial translation and attenuating the valgus loading at the knee joint [46]. By securing initial balance during single-limb landing, the mGas activation plays a vital role in maintaining postural stability and preventing potential knee joint instability or injuries [47].

4.2.2. Age-Specific Muscle Synergy

We found that the activation of the biceps femoris (BF) is a prominent component of the specific muscle synergy observed in older adults. The hamstrings, including the BF, are primarily responsible for hip extension when the knee is extended [48]. Therefore, the prominent activation of the BF during the single-limb stance phase in older adults may serve to facilitate hip extension, counteracting the greater trunk flexion. This finding supports the greater hip extension torque observed in older adults. By reducing trunk flexion through hip extension, older adults can also better prepare for the subsequent protective (swinging) leg compensatory step during balance recovery.

In younger adults, we observed a prominent activation of the gluteus medius (Gmed) in their specific muscle synergy. The Gmed serves as a prime mover for hip joint abduction, which plays a crucial role in controlling the movement of the CoM from the lateral to medial side when the standing (perturbed) leg is fixed on the ground. Following a surface drop perturbation, the CoM shifts laterally, disrupting ML balance as it moves outside of its base of support (BoS) during the initial phase of the SLS phase.

To restore ML balance, it is necessary to bring the CoM back to the center and within the BoS. The activation of Gmed actively contributes to this process by controlling the displacement of the CoM from the lateral to medial side, directing it towards the inside of the BoS [49]. The prominent activation of Gmed in younger adults effectively assists in stabilizing ML balance, as indicated by their reduced postural sway (SDCoMAccel) in the ML direction during the SLS phase. The observation of greater hip abduction torque in younger adults compared to older adults supports this activation pattern of the Gmed specific muscle synergy. By generating greater hip abduction torque through Gmed activation, younger adults are better able to counteract the lateral displacement of CoM and effectively restore ML balance following unexpected surface drop perturbations.

### 4.3. Clinical Significance

The comparison between older and younger adults in this study emphasizes the importance of appropriate trunk control and its impact on hip abduction and ML balance control. These findings underscore the significance of interventions that specifically target trunk control and promote optimal coordination between trunk and hip controls for balance recovery in older adults.

These findings provide valuable insights into age-related changes in neuromuscular control and movement strategies for balance control, supporting rehabilitation interventions that focus on trunk control and hip abduction/adduction strengthening for older adults. These targeted interventions have the potential to promote optimal trunk and hip coordination in older adults during dynamic balance recovery.

### 4.4. Limitations

There are a few limitations of the present study that should be mentioned. Given our small sample size, the study's results must be interpreted with caution, and future research

with a larger sample size is required to assess the entire older population more accurately. To further verify our findings, more muscles in the upper body and arms which contribute to total postural balance control during the SLS phase will be included in future studies.

## 5. Conclusions

Older adults demonstrated greater trunk flexion and hip extension torque, while younger adults displayed increased hip abduction torque during the SLS phase of balance response. The excessive trunk flexion observed in older adults can disrupt the timing and effectiveness of hip abduction, leading to delayed ML balance response. Greater hip extension in older adults can be considered as a compensatory movement strategy to counteract excessive trunk flexion. However, the combination of increased hip extension and delayed ML balance responses may contribute to increased postural sway in older adults.

**Author Contributions:** Conceptualization, W.J., J.W. and K.W.; methodology, W.J., A.R.; software, W.J., A.R.; validation, W.J., A.R. and K.W.; formal analysis, W.J.; investigation, W.J.; resources, W.J., A.R.; data curation, W.J., N.A.; writing—original draft preparation, W.J., A.R., N.A., J.W. and K.W.; writing—review and editing, W.J., A.R., N.A., J.W. and K.W.; visualization, W.J.; supervision, W.J., J.W., N.A. and K.W.; funding acquisition, W.J. All authors have read and agreed to the published version of the manuscript.

**Funding:** This research was funded by the University of Maryland Advanced Neuromotor Rehabilitation Research Training (UMANRRT), Program Post-doctoral Fellowship from National Institute on Disability, Independent Living, and Rehabilitation Research (NIDILRR grant number 90AR5028).

**Institutional Review Board Statement:** The study was conducted in accordance with the Declaration of Helsinki and approved by the Institutional Review Board of University of Maryland (protocol code: HP-00093655 and 21 January 2021).

**Informed Consent Statement:** Informed consent was obtained from all subjects involved in the study.

**Data Availability Statement:** Not applicable.

**Conflicts of Interest:** The authors declare no conflict of interest.

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
