# Peer review of "Age-Related Differences in Kinematics, Kinetics, and Muscle Synergy Patterns Following a Sudden Gait Perturbation: Changes in Movement Strategies and Implications for Fall Prevention Rehabilitation"

_applsci, doi:10.3390/app13159035_

Round 1

Reviewer 1 Report

This manuscript entitled “Age-related Differences in Kinematics, Kinetics, and Muscle Synergy Patterns Following a Sudden Gait Perturbation: Changes in Movement Strategies and Implications for Fall Prevention Rehabilitation” primarily aimed to investigate age-related differences in lower limb kinematics, kinetics, and muscle synergy patterns during SLS, as well as their relationship to postural control strategies. To enhance the quality of the manuscript, revise suggestions are given below.

In the introduction, the authors present possible causes of falls in older adults. But the control group for this study was young people. I think adding a literature review of the comparison between the two in this part will make it easier for readers to understand.

Lines 48-52: The ability to successfully recover balance following unexpected ground perturbations depends on how well the forward and downward falling momentum is decelerated in the anterior-posterior (AP) and vertical directions, while simultaneously maintaining postural stability in the medio-lateral (ML) direction. The reviewer agrees with the author’s opinion about this, perhaps the authors can refer to the follow research literature to further verify the opinion: A. Single-Leg Landings Following a Volleyball Spike May Increase the Risk of Anterior Cruciate Ligament Injury More Than Landing on Both-Legs (DOI: 10.3390/app11010130); B. Temporal Kinematic Differences between Forward and Backward Jump-Landing (DOI: 10.3390/ijerph17186669); C. An Investigation of Differences in Lower Extremity Biomechanics During Single-Leg Landing From Height Using Bionic Shoes and Normal Shoes (DOI: 10.3389/fbioe.2021.679123); D. Temporal kinematic and kinetics differences throughout different landing ways following volleyball spike shots (DOI: 10.1177/17543371211009485)

How does the author define young and old?

The subjects selected for this study span a wide range of ages and vary in gender. My question is whether this sample size can support the entire research results if men and women are compared together?

The data acquisition part needs a more precise description, for example, the acquisition frequency of Vicon, the acquisition frequency of EMG and so on. The authors focus a large amount of text on conservation measures. While that's great, the acquisition device is the key piece of equipment for the entire experiment.

Please add limitation

Author Response

Reviewer 1

This manuscript entitled “Age-related Differences in Kinematics, Kinetics, and Muscle Synergy Patterns Following a Sudden Gait Perturbation: Changes in Movement Strategies and Implications for Fall Prevention Rehabilitation” primarily aimed to investigate age-related differences in lower limb kinematics, kinetics, and muscle synergy patterns during SLS, as well as their relationship to postural control strategies. To enhance the quality of the manuscript, revise suggestions are given below.

In the introduction, the authors present possible causes of falls in older adults. But the control group for this study was young people. I think adding a literature review of the comparison between the two in this part will make it easier for readers to understand.

Thanks for this valuable comment. We have added some articles showing the difference between older and younger adults, which demonstrate age-related changes in balance control.

Lines 39-46: “Previous studies demonstrated that age-related changes in neuromuscular control, muscle weakness, and decreased flexibility led to altered muscle activation patterns during balance recovery while standing and walking. For example, compared to younger adults, older adults demonstrated less modulation of spinal reflexes and more co-contraction at the ankle joint for balance control during standing on a compliant surface [4]. A decrease in efficiency of plantar flexion push-off power during standing balance control in response to lateral surface perturbations is also observed in older adults [5].”

Lines 48-52: The ability to successfully recover balance following unexpected ground perturbations depends on how well the forward and downward falling momentum is decelerated in the anterior-posterior (AP) and vertical directions, while simultaneously maintaining postural stability in the medio-lateral (ML) direction. The reviewer agrees with the author’s opinion about this, perhaps the authors can refer to the follow research literature to further verify the opinion: A. Single-Leg Landings Following a Volleyball Spike May Increase the Risk of Anterior Cruciate Ligament Injury More Than Landing on Both-Legs (DOI: 10.3390/app11010130); B. Temporal Kinematic Differences between Forward and Backward Jump-Landing (DOI: 10.3390/ijerph17186669); C. An Investigation of Differences in Lower Extremity Biomechanics During Single-Leg Landing From Height Using Bionic Shoes and Normal Shoes (DOI: 10.3389/fbioe.2021.679123); D. Temporal kinematic and kinetics differences throughout different landing ways following volleyball spike shots (DOI: 10.1177/17543371211009485)

 The recommended references have been added to the text. Thank you!

Lines 53-57: “The ability to successfully recover balance following unexpected ground perturbations depends on how well the forward and downward falling momentum is decelerated in the anterior-posterior (AP) and vertical directions, while simultaneously maintaining postural stability in the medio-lateral (ML) direction [10–13].”

How does the author define young and old?

Thanks for pointing this out. This has been added to “2.1 Participants”

Lines 105-106: “Older adults were defined as individuals over 65 years of age, while younger adults were between 18 to 30 years old.”

The subjects selected for this study span a wide range of ages and vary in gender. My question is whether this sample size can support the entire research results if men and women are compared together? Thanks for this feedback. We acknowledge that a small sample size can limit this study’s validity and overall ability to draw meaningful conclusions. This concern was mentioned in the limitation section (Lines 424-429).

The data acquisition part needs a more precise description, for example, the acquisition frequency of Vicon, the acquisition frequency of EMG and so on. The authors focus a large amount of text on conservation measures. While that's great, the acquisition device is the key piece of equipment for the entire experiment. Thanks for pointing this out. Data acquisition part has been elaborated.

Lines 175 – 184: “The raw surface EMG response to the perturbation was filtered with a 20 - 450 Hz band-pass filter. Subsequently, the EMG data was high-pass filtered at 35 Hz, and a 2nd order Butterworth low pass filter with a 40 Hz cutoff was applied as a digital smoothing algorithm after full wave rectification [24,25]. EMG data were sampled at 1,500 Hz and kinematic data was sampled at 150 Hz. EMG, kinematics, and kinetics data were analyzed in Matlab 2022b (Matworks Inc., Natick, MA, USA).

2.3.1 Kinematic and kinetic data

The body’s center of mass (CoM) trajectory, angular displacement and torque of the ankle and hip joints were calculated using the Plug-in Gait dynamic pipeline (Vicon, Ox-ford Metrics, UK).”

Please add limitation

A limitation section has been added.

      Lines 424-429: “There are a few study limitations that should be mentioned. Given our small sample size, results must be interpreted with caution, and future research with a larger sample size is warranted to assess the entire older population more accurately. To further verify our findings, more muscles in the upper body and arms, which contribute to total postural balance control during the SLS phase, will be included in future studies.”

Reviewer 2 Report

Thank you for the opportunity to review this valuable work.

The theme of this study is interesting and important. The founding of movement strategies or muscle synergies are valuable for preventing falling or management of rehabilitation. This well structured and writed study was also questioned the postural control strategies and demonstrated important findings on this field.

I don't have any comments for major revisions. Congratulations to the authors for their effort.

Introduction

The background of the research was clear. The authors put their originality in the line with literature.

Methods

·         Please add Ethical approval ID number into the method.

·         Please provide more information on the study population. How were patients recruited - the text just says the samples' collection,? It is important to understand the source population for the study.

·         In this cohort, I think the main limitation was the small sample size. The authors should to present sample size calculation.

Results

Results were nicely presented and supported with figures.

Discussion

·         Please support your statements with related references. The number of references are not enough for the discussion.

·         I did not see any limitations of the study, the authors can expand the end of the discussion with limitations.

Author Response

Reviewer 2

Thank you for the opportunity to review this valuable work.

The theme of this study is interesting and important. The founding of movement strategies or muscle synergies are valuable for preventing falling or management of rehabilitation. This well structured and writed study was also questioned the postural control strategies and demonstrated important findings on this field.

I don't have any comments for major revisions. Congratulations to the authors for their effort. Thank you!

Introduction

The background of the research was clear. The authors put their originality in the line with literature.

Methods

  • Please add Ethical approval ID number into the method. Thanks for pointing this out. This has been added.

      Lines 129-132: “All procedures conducted in this study were approved by the Institutional Review Board at the University of Maryland (protocol code: HP-00093655 and 1/21/2021) and were conducted in accordance with the Helsinki Declaration of 1975”

  • Please provide more information on the study population. How were patients recruited - the text just says the samples' collection,? It is important to understand the source population for the study. This has been added to the Method.

      Lines 104-106: “Participants were recruited through a weekly department newsletter, online campus advertisements, and the “ResearchMatch” program website. Older adults were defined as individuals over 65 years of age, while younger adults were between 18 to 30 years old.”

  • In this cohort, I think the main limitation was the small sample size. The authors should to present sample size calculation. Thank you for the valuable comment. The process of sample size calculation was shown in lines 227-231: “A priori power analysis using G*Power were performed for sample size justification. Effect size (Cohen’s d) was determined by calculating the mean difference between two groups based on previous research [15,33,34]. The effect size of 1.154 was detected where there was 80% power at the 0.05 alpha level (two-tailed).”

In addition, a limitation section has been added to address this concern and ensure that the readers can interpret the results in a balanced manner.

      Lines 424-429: “There are a few study limitations that should be mentioned. Given our small sample size, results must be interpreted with caution, and future research with a larger sample size is warranted to assess the entire older population more accurately. To further verify our findings, more muscles in the upper body and arms, which contribute to total postural balance control during the SLS phase, will be included in future studies.”

Results

Results were nicely presented and supported with figures. Thank you!

Discussion

  • Please support your statements with related references. The number of references are not enough for the discussion. We have added related references throughout the discussion. The additions are highlighted in yellow). Thank you!

     Lines 325-328: “An unexpected surface drop leads to a reflex overaction of the back muscles, resulting in greater trunk flexion [37]. This trunk flexion following an unexpected surface drop poses a challenge to the dynamic stability of forward-directed locomotion [38]”

      Lines 340-341: “Trunk flexion alters the alignment and position of the body’s CoM, shifting it forward [43]”

     Lines 352-353: “In addition, increased demand for AP balance control, resulting from the control of greater trunk flexion [45]”

     Lines 358-359: “This L-to-M displacement of the CoM is important in preventing potential loss of balance in the ML direction [46]”

     Lines 383-384 “the mGas activation plays a vital role in maintaining postural stability and preventing potential knee joint instability or injuries [49]”

     Lines 402-404 “The activation of Gmed actively contributes to this process by controlling the displace-ment of the CoM from the lateral to medial side, directing it towards the inside of the BoS [51].”

  • I did not see any limitations of the study, the authors can expand the end of the discussion with limitations. We have incorporated supporting content from previous studies to strengthen our findings. Additionally, a limitation section has been included to address potential shortcomings in our study.

      Lines 325-327: “An unexpected surface drop leads to a reflex overaction of the back muscles, resulting in greater trunk flexion [37]. This trunk flexion following an unexpected surface drop poses a challenge to the dynamic stability of forward-directed locomotion [38].”

      Lines 424-429: “There are a few study limitations that should be mentioned. Given our small sample size, results must be interpreted with caution, and future research with a larger sample size is warranted to assess the entire older population more accurately. To further verify our findings, more muscles in the upper body and arms, which contribute to total postural balance control during the SLS phase, will be included in future studies.”

Reviewer 3 Report

The article – “Age-related Differences in Kinematics, Kinetics, and Muscle Synergy Patterns Following a Sudden Gait Perturbation: Changes in Movement Strategies and Implications for Fall Prevention Rehabilitation” is an interesting experiment carried out to elucidate the internal mechanisms for maintaining balance in the event of an unexpected perturbation.

The authors have succeeded in presenting quite a significant amount of material in a compact way. The Discussion section could be left as such, however, taking into account the practical importance of the data obtained, I would recommend the authors to provide this section with several schematic drawings that could visually (especially for those who are not very involved in the research topic) demonstrate the patterns you have obtained (Common, Older adults specific, Younger adults specific). This refers to the simplest a stick model kinematics at SLS phase. This illustrative picture would help many readers to better understand the patterns you discovered. Yes, it is possible that making such an illustration may be more difficult than it seems, but it definitely matters for the perception of your research. Taking into account the fact that any illustration requires certain efforts, which, given the time allotted by the editors, may not be able to be realized, I do not insist on this.

Except that the Line 119 - Need to fix on BODY Mass Index

Author Response

Reviewer 3

The article – “Age-related Differences in Kinematics, Kinetics, and Muscle Synergy Patterns Following a Sudden Gait Perturbation: Changes in Movement Strategies and Implications for Fall Prevention Rehabilitation” is an interesting experiment carried out to elucidate the internal mechanisms for maintaining balance in the event of an unexpected perturbation.

The authors have succeeded in presenting quite a significant amount of material in a compact way. The Discussion section could be left as such, however, taking into account the practical importance of the data obtained, I would recommend the authors to provide this section with several schematic drawings that could visually (especially for those who are not very involved in the research topic) demonstrate the patterns you have obtained (Common, Older adults specific, Younger adults specific). This refers to the simplest a stick model kinematics at SLS phase. This illustrative picture would help many readers to better understand the patterns you discovered. Yes, it is possible that making such an illustration may be more difficult than it seems, but it definitely matters for the perception of your research. Taking into account the fact that any illustration requires certain efforts, which, given the time allotted by the editors, may not be able to be realized, I do not insist on this.

Thanks for your great feedback. The figure 5 has been revised according to your comment (line296).

Except that the Line 119 - Need to fix on BODY Mass Index

Corrected. Thank you!

Line 127: “4) Body Mass Index (BMI) within the overweight and obesity range”
